# Low-Complexity Nonlinear Self-Inverse Permutation for Creating Physically Clone-Resistant Identities

**Saleh Mulhem ***[ORCID]**, Ayoub Mars**[ORCID] **and Wael Adi**

Institute of Computer and Network Engineering, Technische Universität Braunschweig, 38106 Braunschweig, Germany; a.mars@tu-braunschweig.de (A.M.); w.adi@tu-braunschweig.de (W.A.)

* Correspondence: s.mulhem@tu-bs.de; Tel.: +49-176-238-767-03

**Abstract:** New large classes of permutations over $\mathbb{Z}_{2^n}$ based on T-Functions as Self-Inverting Permutation Functions (SIPFs) are presented. The presented classes exhibit negligible or low complexity when implemented in emerging FPGA technologies. The target use of such functions is in creating the so called Secret Unknown Ciphers (SUC) to serve as resilient Clone-Resistant structures in smart non-volatile Field Programmable Gate Arrays (FPGA) devices. SUCs concepts were proposed a decade ago as digital consistent alternatives to the conventional analog inconsistent Physical Unclonable Functions PUFs. The proposed permutation classes are designed and optimized particularly to use non-consumed Mathblock cores in programmable System-on-Chip (SoC) FPGA devices. Hardware and software complexities for realizing such structures are optimized and evaluated for a sample expected target FPGA technology. The attained security levels of the resulting SUCs are evaluated and shown to be scalable and usable even for post-quantum crypto systems.

**Keywords:** self-inverse permutations; secret unknown cipher; system-on-chip FPGA; physical unclonable functions

## 1. Introduction

The demand for a more efficient and secure physical identification of the participating entities in a security architecture is a permanent technological need. Unclonable physical identities are emerging to play a crucial role in contemporary and future security systems such as those dealing with smart houses, smart cities, smart healthcare, etc. [1]. In the last two decades, many physical identification approaches have been proposed based on simply verifying a stored secret key in an embedded non-volatile memory (NVM) [2]. This approach is inherently cloneable as somebody knows the stored secret keys. moreover, it mostly failed to face simple replacements/physical attacks [3]. A more sophisticated approach was proposed based on "unknown" Physical Unclonable Functions (PUFs) serving as a physically unclonable identity for RFIDs, smart cards, mobile devices, and, generally, the Internet of Things (IoT) devices [2]. PUFs use intrinsic properties of electronic device structures which cannot be manufactured to become identical and is comparable to a borne biological DNA. PUFs properties are fully random, obscure, and unpredictable. Therefore, PUFs are theoretically considered as unknown functions that are impossible to clone [4]. However, all PUFs share the same property of being inconsistent as unknown continuous "analog" mappings.

Our proposed "digital" Secret Unknown Cipher (SUC) concept was introduced a decade ago in [5] as electronic DNA (e-DNA). SUCs physical modules compared to PUFs, are highly consistent as pure digital structures. SUCs are considered as "clone-resistant digital functions" that can be realized as digital structures. SUCs are self-created and embedded within off-shelf System on Chip (SoC) Field Programmable Gate Arrays (FPGA) devices in a post-fabrication process, where the device manufacturer can be excluded from the security process. This work shows a possible approach

towards creating such SUCs in emerging future SoC FPGA technologies. The particular property of the proposed cipher classes for SUC is that they use existing unused-arithmetic-hardcore-modules that are already available in most modern FPGA resources.

The main contributions of this work are in devising a new technique for converting future smart programmable VLSI devices into physically hard-to-clone units as an alternative to conventional PUF technologies. The technique is based on a new concept in creating permanent and highly resilient digital SUC. Therefore, new special huge classes of FPGA-optimized ciphers are introduced based on low-cost self-inverse permutation functions deploying special FPGA arithmetic-hardwired-cores. The targeted VLSI technologies are emerging future smart self-reconfiguring and non-volatile SoC FPGA devices to accommodate the proposed SUC-modules. Such technologies are expected to emerge in the near future. The paper introduces a new unknown cipher generator based on new self-inverse permutations to construct digital SUCs to replace the inconsistent analog PUFs in a large class of applications.

The paper is organized as follows.

- The state of the art of PUFs as recently used unknown functions in Section 2. Other proposals of unknown functions were carefully reviewed in [6].
- In Section 3, the creation process of SUCs is presented in more details to make the paper self-contained.
- Then the basic algebra to be deployed for cipher construction is defined based on an expected future VLSI environment in Section 4.
- In Sections 5–7, the novel found large classes of self-inverse permutations over $\mathbb{Z}_{2^n}$ are introduced.
- In Section 8, the hardware implementation complexities are evaluated.
- Finally, the resulting cipher structures and their attained security levels are discussed in Section 9.
- Section 10 concludes the results.

## 2. Background Motivation and State of the Art on Physical Unclonability

Tremendous research efforts were conducted two decades ago to create unclonable entities. The conventional Physical Unclonable Functions (PUFs) as technologies to make physical units unclonable were introduced in the last two decades such as in [2,7,8]. A PUF was first introduced as a physical one-way function in an optical environment [9]. Then, the PUF was proposed as a controlled physical random function in [10]. Later, the PUF was defined as an unknown function [11]. Several electronic PUFs were presented such as ring oscillator PUFs [10,12], TERO-PUF [13], arbiter PUFs [14], Chaos-based PUFs [15], etc. Furthermore, PUFs were classified into two categories: If a PUF can produce an enormous number of input-output pairs (challenge-response pairs (CR-pairs)), then it is assigned as a Strong PUF. Otherwise, the PUF is assigned as a weak PUF if it produces a limited number of CR-pairs [2].

In the following sections, several proposals deploying PUFs are selected to be reviewed as being related to our work. The following technical discussions cover also PUF-vulnerabilities, as well as modeling attacks, as the most important threats facing PUF technologies.

### 2.1. PUFs Drawbacks and Disadvantages

As nonlinear and unknown analog mappings, most PUFs output responses cannot be perfectly reproducible for the same input stimuli. Such a response inconsistency occurs due to many reasons such as environmental perturbations such as supply voltage, noise, operation temperature, aging, and many other specific effects. The PUF-Drawbacks can be classified as follows [2]:

1.  Inconsistent input-output behavior and consequently inconsistency in the PUF's CR-pairs reproducibility.

    In addition to two mainly security drawbacks, namely:

2.  Possible correlations in PUF CR-pairs allowing modeling attacks.
3.  A limited or small number of possible distinct CR-pairs which simplifies cloning attacks.

## 2.2. Counteracting the Drawbacks of PUFs

Several PUF-improvements were presented to overcome PUF-drawbacks to make them more robust, reliable and secure. Such improvements were conducted in [16–20] based on error-correction techniques and/or combining PUFs with additional cryptographic primitives.

To counteract PUF-inconsistency drawbacks, complex fuzzy extractors were introduced as a remedy [21]. Such fuzzy extractors deploy Helper Data Algorithm (HDA) and/or error correction code (ECC) procedures to stabilize the PUF's inconsistent responses. As a result, the need for a fuzzy extractor makes most of the proposed PUFs very costly to implement with more latency counteracting most requirements of a lightweight design. For instance, the proposed fuzzy extractor of [20] requires 5.1 times of the PUF's resources to produce more consistent responses.

Correlation drawbacks allow several security threats against PUF technology. Recently, modeling attacks were considered among the most important PUF-threats. The highly correlated PUF CR-pairs lead to building predictive models of the PUFs [22–25]. The general methods of modeling attacks on PUFs can be concluded as follows:

- **Modeling Attack using Machine Learning (ML) Algorithms:** New PUF attacks based on modeling by machine learning (ML) were found as in [22] with alarming high prediction rates approaching 99%. The predictive models using ML techniques of various proposals of delay PUF are constructed with error ratios [22]: less than 1% for Arbiter PUFs, 1% for XOR Arbiter PUFs, 4.5% for Feed-Forward Arbiter PUFs, and less than 1% for Ring Oscillator PUFs.
- **Modeling Attack using PUF-Codebook:** Several PUFs generates a limited number of CRPs [2]. For instance, Ring Oscillator PUFs produce a quadratic number of CRPs. Therefore, an adversary can produce a codebook of the PUF containing a look-up table of all CRPs to imitate the PUF [23].

For instance and in reference to Figure 1a, a controlled PUF was proposed by combining a random hash function with a weak PUF to prevent modeling attacks in [10]. As also HDA is still required, the resulting overall high complexity is mostly not acceptable for practical use.

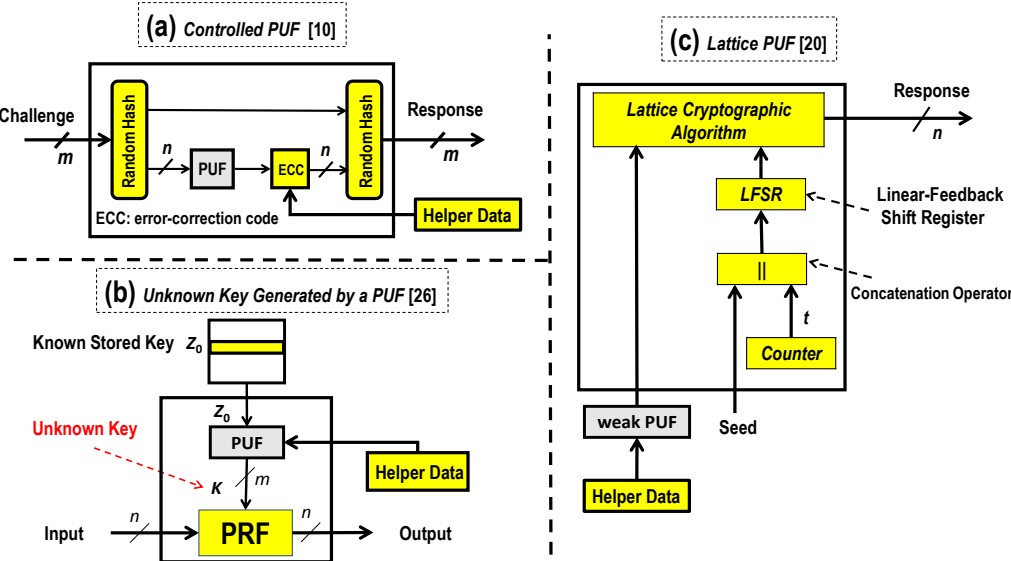

**Figure 1.** Related Physical Unclonable Functions (PUF) - Improvements to Resist Modeling Attacks. Adapted from [10,20,26]. PRF = Pseudorandom function; LFSR = linear-feedback shift register; ECC = error correction code.

In reference to Figure 1b and according to [26], a PUF-based unknown key generation was proposed. The proposed mapping structure uses a PUF as a key generator and Pseudorandom Function (PRF) as input-output mapping resulting with large usable operational space with no correlation and

large usable distinct pairs. Again, complex HDA is required resulting with overall non-attractive high-complexity for practical use.

In reference to Figure 1c and according to [20] using a weak PUF with a lattice cryptographic algorithm and a linear-feedback shift register (LFSR) combined to construct a strong PUF. The LFSR-connection polynomial is used to expand the PUF's CR-space, where the lattice mapping counteracts machine learning attacks.

## 2.3. PUFs Use-Cases and Their Weaknesses

Several PUF- based authentication protocols have been presented. For instance, a delay PUF was deployed as a key generator in an anonymous e-money transaction protocol [27]. In [28], PUF is proposed for a RFID-authentication protocol. Here, the designed protocol requires an extra hash function and the proposed PUF needs to exchange extra data with the server to attain a stable response. In [7], resource-constrained Internet of things (IoT) devices deploying PUF were studied and analyzed. PUFs were introduced as a lightweight solution to ensure secure communications among IoT devices. Unfortunately, the results showed that most of PUF proposals need at least an extra cryptographic algorithm or secure memory to perform the authentication protocols. This need for cryptographic primitives increases the required energy and computational resources for performing PUF-based authentication protocols. In [2], nineteen PUF-based authentication protocols have been studied and analyzed. The results showed that most PUFs are not resilient against modeling attacks, most PUF CR-space requires expanding to be sufficiently large for the entity authentication, and CR-pairs management is very costly due to collision-prone properties of all PUF-mappings.

## 2.4. Recent Alternatives to Analog PUFs

The SUC concept as an alternative full-digital substitute to the analog PUF was proposed by the authors a decade ago in the public literature as in [5,11]. SUCs, as pure digital structures are ultimately consistent [29]. Additionally, SUCs as designed invertible PRFs offer full usage of the whole $2^n$ input-output pairs as plaintext-ciphertext pairs, where, $n$ is the cipher input size.

A possible SUCs creation scenario as Feistel-like ciphers was presented recently in [6]. Involutive SUC and non-involutive SUC classes were proposed in [30]. In [31], a new family of stream ciphers deploying non-linear feedback shift registers was proposed as a possible SUC class. In [32], SUC as a random stream cipher based on T-function was proposed. The current paper is presenting new generalized large classes of useful special low-cost self-inverse permutation mappings by deploying pure FPGA resources as arithmetic-cores based on T-functions for SUC constructions.

To show in a very simplified way the differences of our proposal to the somehow digitized analog PUFs with some relation to the SUC concept, the following two examples are depicted from the public literature.

**Example 1.** *In Figure 2a, the concept of a PUF-based unknown key generation was proposed to construct a strong PUF. In particular, a weak SRAM PUF is used as a key generator for a hardwired (AES) Advanced Encryption Standard cipher [19]. Note that the resulting virtual PUF's input-output mapping is invertible. Therefore, it produces $2^n$ input-output pairs as plaintext-ciphertext pairs, where, n is the cipher input size. Again, HDA or equivalent techniques are still required to stabilize the noisy responses of the used SRAM PUF. The overall hardware complexity results with quite high complexity [19]. In comparison, our SUC solution on the right-side substitutes all required components with unknown digital structures. No complex fuzzy extractor with limited consistency is required. Therefore, the proposed SUC-structures meet the lightweight design requirements for mass products.*

## State of the Art  PUF Constellations Comparable to our Solution

**Figure 2.** Two Proposals Deploying PUFs Comparable to our Solution Based on SUC Technique.

**Example 2.** *Figure 2b shows a proposed Feistel-like block cipher structure deploying PUFs to create an unknown f function [33]. Note that a block cipher deploying PUFs to create f is equivalent to a block cipher with a secret unknown mapping allocated in the 4 x 4-bit SRAM-PUFs are utilized as unknown confusion functions. The resulting cipher is PRF with a fully usable input-output space, however, again is highly complex due to the use of HDA. The comparable SUC-structure on the right side is equivalent to our proposed SUC alternative as a key alternating cipher using unknown new T-function based arithmetic involutions as round functions as would be shown later on.*

Notice that both examples are still using inconsistent analog PUFs with all the above-mentioned drawbacks and disadvantages.

## 3. The SUC Concept and Technology Background

The target of this section is to make the paper self-contained and better understandable for the reader, as SUC concept is not widely known in the public literature. This section is a slightly-modified copy of the Section 3 from our earlier publication [6] on SUC design techniques.

The unknown cipher concept is an entirely new security paradigm in the public literature. The unknown cipher here does not deal with protecting the communications or the links between at least two parties, as a sender and a receiver, which requires the cipher to be commonly known to both parties (Kerckhoffs's principle). In particular, the SUC is fundamentally designed for the identification process to serve as a clone-resistant identity [5]. We postulate that "unclonability" is only possible if unknown structures are created. Therefore, a cipher designed to be embedded as a structure that is unknown to anybody (including its designer) does not violate Kerckhoffs's principle. On the other

hand, SUC should not be confused with "security by obscurity", where the cipher is designed by a cryptographer, known to the manufacturer, and then kept secret and obscure.

SUC creation is a very challenging task. Figure 3 illustrates a possible SUC creation concept in a non-volatile (NV) FPGA device having internal self-reconfiguration capability. A large class of ciphers $\{E_1, E_2 \ldots E_\sigma\}$ are first created such that $\sigma \to \infty$ and offered for selection. Then, a single-event process triggers the FPGA-internal true random number generator (TRNG), leading to select randomly an unknown cipher choice $E_j$ from the infinite number $\sigma$ of the created distinct ciphers. A TRNG hardware module is offered in virtually all modern FPGA devices fulfilling the NIST state of the art standard cryptographic requirements (see TRNG-module specifications in the used FPGA [34] in our proposed prototyping). After this process, all the dashed entities in Figure 3 are then irreversibly killed and fully removed from the chip. The self-reconfiguration in the chip is then irreversibly locked (by a flash bit or fuse) to prohibit any repetition of that single-event SUC creation process. That is the created SUC is not more removable or changeable forever like a DNA. This concept was described intensively in the last decade in our old publications [5,29,35].

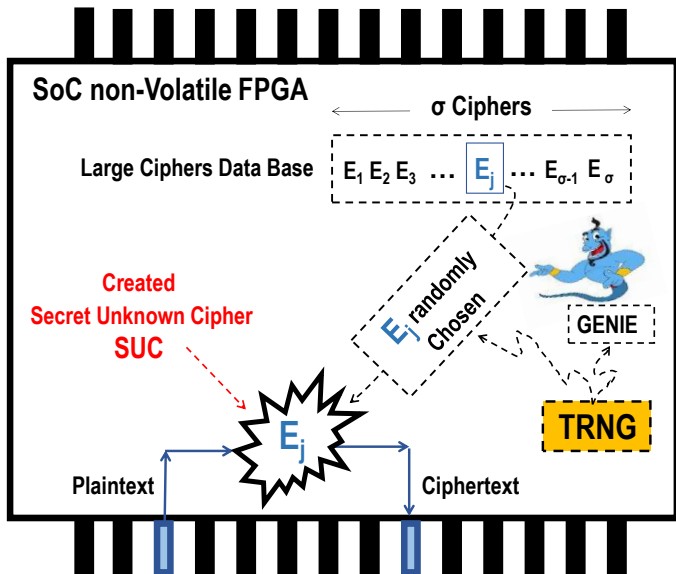

**Figure 3.** Key idea for generating a Secret Unknown Cipher (SUC) [6]. FPGA = Field Programmable Gate Arrays; TRNG = true random number generator.

The resulting cipher is a secret, yet unknown, cipher and is a non-repeatable selection. It is even an unknown choice to the cipher designer/creator himself. The "Secret Unknown Cipher" (SUC) is realizable in an emerging VLSI device that allows self-creation of permanent unknown usable secret structures as "an electronic mutation", as indicated in [36]. Note that for the functionality of the concept, there is no need to publish the SUC creation procedure/program of the cipher class, which is designated from now on as the "GENIE" as a smart cipher designer. However, for worst-case security analysis, we assume that the cipher creating "GENIE" is published.

### 3.1. Creation Concept of Unknown Ciphers as Clone-Resistant Entities/Modules

The proposed SUC is conceptually based on the following principle: "The only secret which can be kept unrevealed is the one which nobody knows" [37]. From a practical point of view, if the cipher creator itself cannot predict and foretell exactly the created cipher, then the cipher is considered as not known when the cipher class size $\sigma \to \infty$.

Figure 4 illustrates a possible SUC creation that is assumed to be processed in a secure environment. The process may proceed as follows:

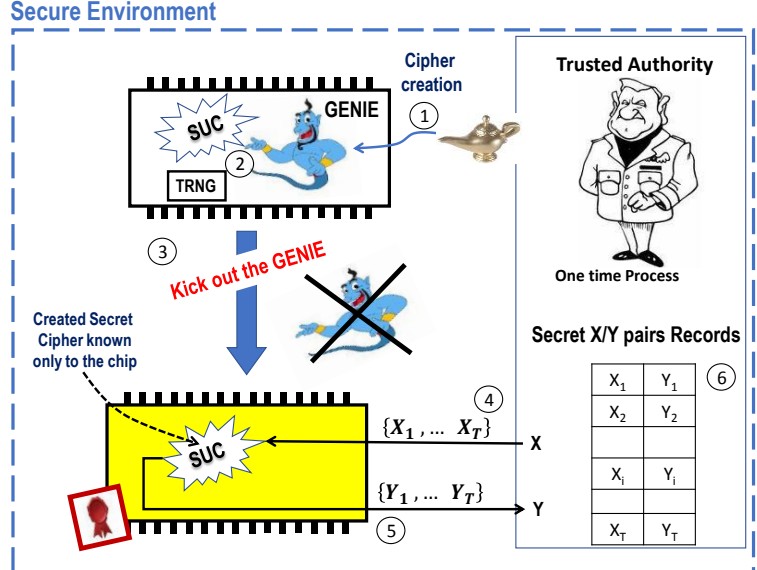

**Figure 4.** Mutating a Secret Unknown Cipher (SUC) into a system-on-chip (SoC) device [6].

**SUC creation phase:**

1. A trusted authority (TA) injects one-time into a system-on-chip (SoC) device the software package "GENIE" as an SUC creator for a short time (as much time as required to create an unknown cipher, which is usually a few milliseconds).

2. Then, the GENIE is internally triggered to generate/select a permanent and unpredictable secure cipher with the help of an internal, non-repeatable, unpredictable, and unknown bit stream from the in-chip TRNG.

3. After creating an SUC, the GENIE is completely and irreversibly deleted. What remains is a non-removable, unchangeable and unknown operational cipher (as SUC) that nobody knows.

**SUC personalization phase:**

4. TA randomly selects a set $\{x_1, \ldots x_T\}$ of cleartext vectors out of the $2^n$ possible combinations, where $n$ is the size of the SUC input/output space in bits.

5. TA stimulates the SoC device to encipher the cleartext vectors into the ciphertexts $\{y_1, \ldots y_T\}$ using its SUC within the device.

6. The resulting $T$-$(x_i, y_i)$ pairs are stored as secret pairs in the individual (personal) device records by the TA. The records should be kept secret for later use.

As the created TRNG bits are fully and exclusively responsibly for creating the SUC, and as TRNG bits are unpredictable, non-repeatable, and unknown, the resulting created SUC in the SoC device is also unknown and unpredictable, such that:

$$SUC_t = GENIE(TRNG_t). \tag{1}$$

Every $t > 0$. This implies that,

$$SUC_t : \{0,1\}^n \times \{0,1\}^{k_t} \rightarrow \{0,1\}^n, \tag{2}$$

where $n$ is the bit size of the SUC input/output space and $k_t$ is the bit size of the cipher's secret key. Thus, the maximum number of distinct possible permutations is $\sigma < 2^n!$ as $2^n!$ is the number of all possible $\{0,1\}^n$ to $\{0,1\}^n$ permutations. Therefore, in that case the number of possible selectable block ciphers of block size $n$ is,

$$\sigma_n < 2^n!. \tag{3}$$

In addition, SUC has the property of being able to generate a large number of distinct CRPs as cleartext/ciphertext pairs, which is upper bounded by $2^n$. This counteracts the lack of CR space in the case of traditional analog PUFs.

The created cipher $SUC_t$ is a result of the $TRNG_t$ random sequence that is not known to anyone. Moreover, it is highly probable that for any two-time points $t_1$ and $t_2$,

$$TRNG_{t_1} \neq TRNG_{t_2} \rightarrow SUC_{t_1} \neq SUC_{t_2}. \tag{4}$$

Therefore, each resulting SoC device has its individual SUC with a probability $\left(1 - \frac{g}{1}\right) \rightarrow 1$.

**How to Use an SUC?**

Figure 5 shows a generic two-way identification protocol using such SUCs for authenticating a personalized SoC$_A$ device.

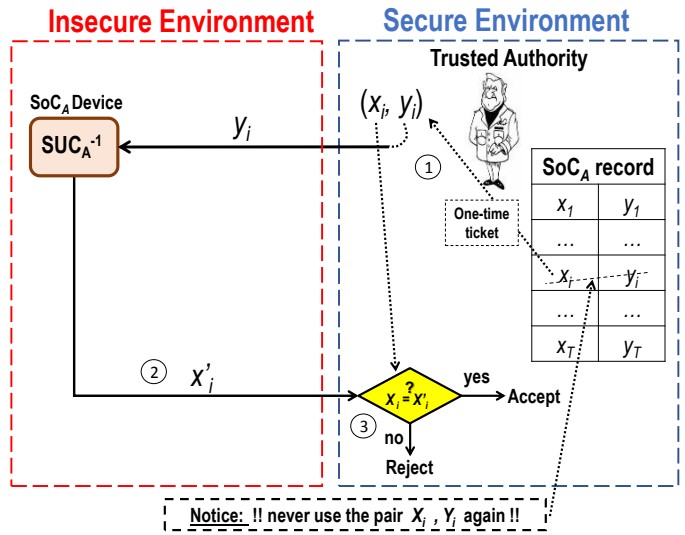

**Figure 5.** Two-way identification protocol over an insecure channel [6].

An SUC-based identification protocol may proceed as follows:

1. A secret pair $(x_i, y_i)$ is randomly chosen from the TA's secret records of SoC$_A$. Then, the TA challenges the SoC$_A$ device by the cryptogram $y_i$ over an insecure channel.
2. The SoC$_A$ device responds by sending the decrypted cleartext $x'_i$.
3. If $x'_i = x_i$, then the SoC$_A$ device is deemed to be authentic, and the pair $(x_i, y_i)$ is then marked as a used pair and never used again avoiding replay attack for highest security.

Refined versions of that protocol are further developed as shown in [38]. It is shown that much more efficient and low-cost CR-pairs management is possible due to the invertibility property of the SUC (with no collision as a cipher) compared to the collision-prone properties of all PUF-mappings (as somehow unknown hash-functions).

*3.2. SUC Application Spectrum*

The concept of SUCs as clone-resistant entity offers new attractive spectrum of applications. In [39], basic identification and usage scenarios are presented. In [40], a use case for securing IP-cores in FPGA environment is introduced. Recently, the authors published several IoT use cases for SUCs attaining very efficient management for secured remote sensing [41]. A Fleet Management System (FMS) for secured logistic operations is presented in [42]. In [43], a novel concept for electronic wallets for e-cash is presented.

## 4. Targeted SUCs Realization in Non-Volatile VLSI-FPGA Environment

Microsemi is the only provider of non-volatile FPGA technology with flash-based distributed switching fabrics and programmable cells. One of the advanced non-volatile SoC FPGAs from this company is SmartFusion®2. The flash-based FPGA fabric incorporates an integrated ARM Cortex-M3 processor together with powerful arithmetic units MACC and high-performance communication interfaces all integrated in a single chip. The infrastructure is smart enough to allow a GENIE program to create a cipher within the SoC unit. However, self-reconfiguration and irreversible reconfiguration-locking is still not possible in such non-volatile devices to enable self-creation of permanent unknown "hard-wired" structures as SUCs. Enabling self-reconfiguration is available in RAM-based FPGA technologies are expected to emerge similarly in the flash-based non-volatile technology in the near future. Assuming that self-reconfiguration would be possible in the future, designers can devise mechanisms for SUC-creation in such devices.

The greatest technology challenges in self-creating of SUCs can be summarized in the following two categories:

1. Designing a GENIE program as a "smart VLSI-designer" which can extend an existing FPGA design without violating the technology design rules.
2. Designing a GENIE which can serve as an obedient "smart cipher creator" to fulfill all necessary security requirements to come-up with a really unknown and unpredictable cipher (SUC).

The above two challenge-categories are technically highly hard to be practically-realized in an ultimate way. However, there are no technical reasons to believe that the SUC creation as a concept may become an impossible mission. Our proposed cipher concept in this paper shows one possible concept targeting to address both challenges in a promising and practically realizable approach.

The main objective of this work was to approach strategies towards making the targets technically possible at low area cost and low GENIE-complexity in time and memory. The particularly selected strategies in this work are:

1. Using unconsumed FPGA resources. In this case the p hardwired arithmetic addition and multiplication cores.
2. To optimally deploy the technology resources in hiding the SUC structure's keys and function parameters.

Figure 6 shows a possible scenario for an incremental embodiment "mutation" of an SUC in Microsemi SmartFusion®2 SoC FPGA. In this scenario, the GENIE should use only resources outside the "functional HW-Cores". That is use mainly not used "free Mathblocks" resources, "free FPGA fabric" and "free NV memory". The result is a distributed SUC hardware and software components at possibly individual locations in each mutated device.

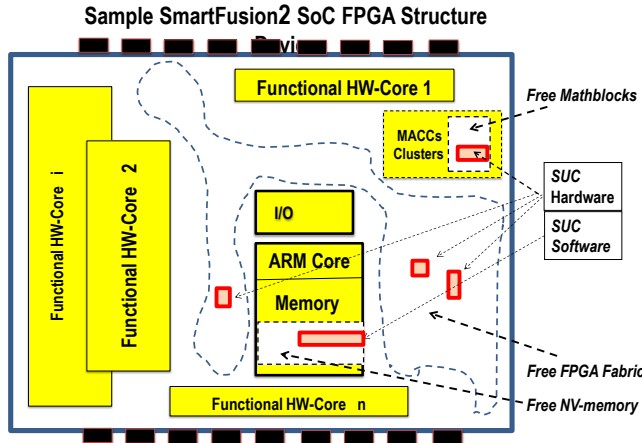

**Figure 6.** Sample functional layout after creating an SUC in an FPGA device [6].

The existing DSP blocks, also known as Mathblocks (MACCs), include multipliers are optimized for example to be efficiently configured to perform $18 \times 18$ or double $9 \times 9$ multiplications allowing implementation of scalable arithmetic. To avoid complex modular arithmetic, the ring of integers arithmetic modulo $2^n$ is adopted as the basic cipher algebra in $\mathbb{Z}_{2^n}$.

Each "Mathblock" offers the following capabilities [34]:

- $18 \times 18$ or double $9 \times 9$ signed multiplications (such as A [17:0] $\times$ B [17:0] or A [17:9] $\times$ B [17:9] + A [8:0] $\times$ B [8:0]).
- Supporting dot product; the multiplier computes: (A [8:0] $\times$ B [17:9] + A [17:9] $\times$ B [8:0]) $\times$ 29.
- Built-in addition, subtraction, and accumulation units to combine multiplication results efficiently.

A multi-precision arithmetic can even be realized in hardware to reach easily a cipher block size $n$ of 64 bits.

Finally, the SoC units should able to be irreversibly prohibit reconfiguration after creating SUC by irreversibly setting some last-fuse hardware-lock.

Based on these existing FPGA resources, the authors consider the cases where many real applications do not consume all the available arithmetic cores. This work is, therefore seeking ciphering classes using mainly simple multiply, add and subtract arithmetic in $\mathbb{Z}_{2^n}$.

The requirements on the cipher design are therefore consequently:

1. Designing huge classes of self-inverse permutation functions modulo $2^n$ using multiplication as a major ciphering function.
2. Creating cascades of such permutations to create powerful (SUCs) to serve as "SUC-PUFs".

## 5. Preliminaries on Crypto-Permutations

This section contains a short introduction to permutations for cipher design. Basic mathematical concepts and lemmas with proofs are introduced to be used throughout this work.

### 5.1. Early Work on Permutation for Ciphering Stages

In cryptography, it is known that confusion refers to providing high degree of non-linearity between plaintext and ciphertext of a cipher, while diffusion refers to the effect of each plaintext- and key-bit on each bit of the ciphertext. Furthermore, diffusion and confusion are considered as the two essential ciphering requirements that inhibit statistical analysis of the input-output pairs behavior. However, a substitution box (S-Box) is usually deployed as a confusion component in the modern cipher. The diffusion component (layer) is widely deployed in a different form as permutation mapping. For instance, a diffusion component is implemented as a fixed wired permutation [44] in designing ultra-lightweight cipher, so-called (PRESENT). In [45], a diffusion wired is configured as a permutation of eight 4-bit words to the design a light-weight cipher. In addition, special matrices were also used as permutation functions in the diffusion layer such as AES cipher [46]. Recently, Yunwen et al. [47] proposed two types of nonlinear functions as alternative diffusing components. The first type relies on a nonlinear code, which is known as a Kerdock code and the second type relies on the T-functions.

On the other hands, Rivest [48] introduced the exact conditions required to find permutation polynomials modulo $2^n$. Moreover, Singh et al. [49] generalized the conditions for permutation polynomials over $\mathbb{Z}_{p^n}$, where p is a prime integer. In [50], the essential conditions for creating self-inverse permutation polynomials of degree 2 over the ring were determined. Another application of permutation polynomial that investigates the permutation polynomial-based interleaves over integer rings for turbo codes is presented in [51]. A quadratic permutation polynomial and its inverse over $\mathbb{Z}_{2^n}$ is presented in [52,53]. Furthermore, an interesting work by Klimov [54,55] introduced the bit-slice analysis method to create the so-called T-functions. Using this method, Klimov et al. [54] proposed more generalized polynomial structures with integer coefficients using (+), (-) and the Boolean Operators over $2^n$. They obtained the necessary and sufficient conditions for such polynomial structures that produce permutation functions.

### 5.2. Permutations

Let $R$ be a finite ring. It is well known that not every function $f : R \to R$ is representable by a polynomial over the same ring. A function $f$ over $R$ is called a polynomial function if there is a polynomial $P(x) \in R[x]$ such that,

$$f(x) = P(x), \ \forall x \in R \tag{5}$$

If a polynomial function $f$ is a one-to-one mapping, then it is called a permutation polynomial (PP) over $R$.

Emerging and modern technologies perform computations modulo $2^n$ efficiently, where, $n = 32, 64$ or 128; therefore, studying permutation polynomials modulo $2^n$ for some integer $n$ would be practical area of interest. In [48], Rivest presented the exact characterization of such PPs modulo $2^n$: $n > 1$.

**Theorem 1.** *A polynomial $P(x) = a_0 + a_1 x + \cdots + a_d x^d$ with integral coefficients is a permutation polynomial modulo $2^n$: $n > 1$, if and only if $a_1$ is odd, $(a_2 + a_4 + \cdots)$ is even, and $(a_3 + a_5 + \cdots)$ is even [48].*

For instance, a polynomial $P(x) = ax + bx^2$ is a Quadratic Permutation Polynomial (QPP) over $\mathbb{Z}_{2^n}$, if $a$ is odd and $b$ is even.

**Definition 1.** *Self-inverse function or an involution is a function $f$ that its own inverse for all $x$,*

$$f(f(x)) = x \tag{6}$$

The following theorem gives the conditions for QPP to be Self-inverse QPP.

**Theorem 2.** *Let $n > 2$ be an integer and let $P(x) = ax + bx^2$ be a QPP over $\mathbb{Z}_{2^n}$. $P(x)$ is self-inverse if and only if the following conditions hold [50]:*

1. *$a = -1 + 2^{n-1} \cdot u$, where $u$ is a unit in $\mathbb{Z}_{2^n}$.*
2. *If $n$ is even, then $b = 2^r \cdot v$, where: $r \geq n/2$ and $v$ a unit in $\mathbb{Z}_{2^n}$.*
3. *If $n$ is odd, then $b = 2^r v$, where: $r \geq (n-1)/2$ and $v$ a unit in $\mathbb{Z}_{2^n}$.*

In cryptographic applications, self-inverse permutations (Involutions) are preferably used in the rounds of block ciphers to attain low implementation complexity. Unfortunately, each permutation polynomial from the previous class does not have the required strong cryptographic properties. In fact, there is a weakness in that at least two fixed points exist in the resulting permutation for every self-inverse permutation polynomial over $\mathbb{Z}_{2^n}$. This weakness of fixed points of the previous class of self-inverse polynomials is proved in the following lemma:

**Lemma 1.** *Any $P(x) = ax + bx^2$ satisfying the conditions of Theorem 2, has exactly two fixed points.*

**Proof.** It is very clear that $x = 0$ is a first fixed point for every self-inverse permutation polynomial of the previous class.

Let $n$ be an even number (odd cases can be proved similarly). For $x = 2^{n-1}$,

$$P(2^{n-1}) = a(2^{n-1}) + b(2^{n-1})^2$$

So,

$$P(2^{n-1}) = (-1 + 2^{n-1}u)(2^{n-1}) + (2^r v)(2^{n-1})^2$$

where $r \geq \frac{n}{2}$ and $u, v$ are units in $\mathbb{Z}_{2^n}$. Thus,

$$P(2^{n-1}) = -2^{n-1} + 2^{2n-2}u + 2^{2n+r-2}v$$

Applying mod $2^n$ on both sides results in,

$$P(2^{n-1}) = 2^n - 2^{n-1} = 2^{n-1}(2-1) = 2^{n-1}$$

which proves that $x = 2^{n-1}$ is a second fixed point.

To prove that every self-inverse permutation polynomial of the previous class has no more than the two fixed points x = 0,$2^{n-1}$, suppose that $x_0 \neq 0,2^{n-1}$ is a fixed point. Then,

$$P(x_0) = a(x_0) + b(x_0)^2 = x_0$$

After substitution,

$$(-1 + 2^{n-1}u)x_0 + (2^r v)x_0{}^2 = x_0$$

Thus,

$$2^{n-1}u \cdot x_0 + 2^r v \cdot x_0^2 = 2x_0$$

Multiplying both sides by 2 results with,

$$2^n u \cdot x_0 + 2^{r+1}v \cdot x_0^2 = 2^2 x_0$$

or,

$$2^n u \cdot x_0 + 2(2^r v \cdot x_0 - 2)x_0 = 0$$

Applying mod $2^n$ on both sides results in,

$$(2^r v \cdot x_0 - 2)x_0 = 0$$

This is only possible if $x_0 = 0$ or $x_0 = (2^{r-1}v)^{-1} \bmod 2^n$, however $2^{r-1}v$ is not an invertible element over $\mathbb{Z}_{2^n}$. Therefore, $x_0 = 0$ contradicts with the assumption $x_0 \neq 0,2^{n-1}$. As a result, every self-inverse permutation polynomial of the previous class has no more than just the above given two fixed points x = 0,$2^{n-1}$, Q.E.D. □

### 5.3. T-Function Principles

In [54], Klimov and Shamir introduced a new class of low-complexity functions, so-called Triangular-Functions (T-function), which are invertible and exhibiting special cryptographic properties.

For $x \in \mathbb{Z}_{2^n}$, a binary representation of $x$ as $n$-bit vectors is given as follows,

$$x = [x]_{n-1} \cdots [x]_1 [x]_0 \text{ where, } [x]_i \in \mathbb{Z}_2 \text{ for every } 0 \leq i < n$$

**Definition 2.** *A function $f(x)$ from an n-bit input to an n-bit output with the property that the $i^{\text{th}}$ bit of its output depends only on the first, the second ... and the $i^{\text{th}}$ bit of its inputs is called a T-function (short for triangular function).*

*Eight basic possible constructing operations of T-functions were introduced as [52]:*

- *Negation $(-a) \bmod 2^n$, Addition $(a+b) \bmod 2^n$.*
- *Subtraction $(a-b) \bmod 2^n$, Multiplication $(a \cdot b) \bmod 2^n$, and.*
- *The Boolean functions; Complement $\bar{a}$, OR $(a \vee b)$, and $(a \wedge b)$, and XOR $(a \oplus b)$,*

   *where, a and b are two n-bit words.*

The following lemma gives an abstract bit-slicing representation of the arithmetic and logic operations,

**Lemma 2.** *For $i > 0$, and $x, y \in \mathbb{Z}_{2^n}$,*

$$
\left.
\begin{aligned}
[x \times y]_0 &= [x]_0 \wedge [y]_0, \quad [x \times y]_i = [x]_i \alpha_{[y]_0} \oplus [y]_i \alpha_{[x]_0} \oplus \alpha_{x \times y} \\
[x \pm y]_0 &= [x]_0 \oplus [y]_0, \quad [x \times y]_i = [x]_i \oplus [y]_i \oplus \alpha_{x \pm y} \\
[x \oplus y]_0 &= [x]_0 \oplus [y]_0, \quad [x \oplus y]_i = [x]_i \oplus [y]_i \\
\left[ x \overset{\wedge}{_\vee} y \right]_0 &= [x]_0 \overset{\wedge}{_\vee} [y]_0, \quad \left[ x \overset{\wedge}{_\vee} y \right]_i = [x]_i \overset{\wedge}{_\vee} [y]_i
\end{aligned}
\right\},
\tag{7}
$$

*where, $\alpha$'s denotes a parameter [56].*

In [54], Klimov generalized Rivest's construction of PPs resulting with invertible mappings with T-functions properties as follows:

**Theorem 3.** *Let $P(x) = a_0 \overset{\oplus}{_+} a_1 x \overset{\oplus}{_+} \cdots \overset{\oplus}{_+} a_d x^d$ be a generalized polynomial with integral coefficients. Then $P(x)$ defines a permutation polynomial modulo $2n$: $n > 2$, if and only if $a_1$ is odd, $(a_2 + a_4 + \cdots)$ is even, and $(a_3 + a_5 + \cdots)$ is even.*

**Definition 3.** *"A T-function $f(x)$ is invertible if it can be represented in the following form:*

$$
[f(x)]_i = [x]_i + \alpha
$$

*which holds for any bit $i$ and $\alpha$ is a parameter" [56]. Finding the inverse of a T-function is not straightforward and not known in general form.*

*The main target of the subsequent sections is to construct T-functions with SIQPF properties deploying the operations $(+), (-),$ and $\oplus$. Therefore, it is necessary to determine the required cryptographically relevant conditions on $P(x) = ax \overset{\oplus}{_+} bx^2$ or $P(x) = a \overset{\oplus}{_+} bx \overset{\oplus}{_+} cx^2$, such that $P(x)$ becomes a Self-Inverse Quadratic Permutation Function (SIQPF) over the ring $\mathbb{Z}_{2^n}$ and $P(x)$ exhibit different fixed points for different permutations.*

## 6. New Classes of Self-Inverse Permutations

In the following, it is shown how to construct QPPs deploying +, -, and $\oplus$ operations to come up with SIQPFs.

**Definition 4.** *The functions*

$$
P(x) = ax \overset{\oplus}{_+} bx^2,
\tag{8}
$$

*or*

$$
P(x) = a \overset{\oplus}{_+} bx \overset{\oplus}{_+} cx^2,
\tag{9}
$$

*are said to be Quadratic Permutation Functions (QPFs) over $\mathbb{Z}_{2^n}$, if they permute all the elements of $\mathbb{Z}_{2^n}$.*

**Definition 5.** *A self-inverse function is a function $f$ over $\mathbb{Z}_{2^n}$, such that,*

$$
f(f(x)) \bmod 2^n = x,
\tag{10}
$$

*where $x$ is an n-bit word.*

*Choosing the coefficients a, b according to Theorem 2, a, b can be represented as n-bit vectors as follows:*

$$
a = [a]_{n-1} \cdots [a]_1 [a]_0 = 01 \underbrace{\cdots 1}_{n-1} \text{ and } b = \genfrac{}{}{0pt}{}{1}{0} \cdots \genfrac{}{}{0pt}{}{1}{0} \underbrace{0 \cdots 0}_{r},
\tag{11}
$$

*where $r \geq \frac{n}{2}$ if n is even and $r \geq \frac{n-1}{2}$ if n is odd.*

**Lemma 3.** *For n >2. A polynomial $P(x) = a \overset{\oplus}{+} bx \overset{\oplus}{+} cx^2$ is a permutation polynomial over $\mathbb{Z}_{2^n}$ if and only if b is odd and c is even (according to Theorem 3).*

**Lemma 4.** *In $\mathbb{Z}_{2^n}$:*

- *If $a = 01 \underbrace{\cdots 1}_{n-1}$, then $a^2 \bmod 2^n = \underbrace{0 \cdots 0}_{n-1} 1 = 1$, and $(a+1) \bmod 2^n = 1\underbrace{0\cdots 0}_{n-1} = 2^{n-1}$.*

- *If $a = 1\underbrace{1\cdots 1}_{n}$, then $a^2 \bmod 2^n = \underbrace{0\cdots 0}_{n-1}1 = 1$, $(a+1) \bmod 2^n = 0$ and $(a.u) \bmod 2^n = 2^n - u$*

- *If $u = \begin{smallmatrix}1\\0\end{smallmatrix} \cdots \begin{smallmatrix}1\\0\end{smallmatrix} \underbrace{0\cdots 0}_{i}$ and $v = \begin{smallmatrix}1\\0\end{smallmatrix} \cdots \begin{smallmatrix}1\\0\end{smallmatrix} \underbrace{0\cdots 0}_{j}$ are two even numbers, then $u \cdot v = \begin{smallmatrix}1\\0\end{smallmatrix} \cdots \begin{smallmatrix}1\\0\end{smallmatrix} \underbrace{0\cdots 0}_{i+j}$, where $i, j \leq n$.*

- *If $b = \begin{smallmatrix}1\\0\end{smallmatrix} \cdots \begin{smallmatrix}1\\0\end{smallmatrix} \underbrace{0\cdots 0}_{r}$ is an even number, then $b^k \bmod 2^n = \underbrace{0\cdots 0}_{n} = 0$ for $k \cdot r \geq n$.*

The proof is very simple. It is based on the definitions of the multiplication and the modulus over $\mathbb{Z}_{2^n}$. Based on the previous lemma, the following theorem is proved:

**Theorem 4.** *: Let n > 2 be an integer. The permutation function $P(x) = ax \overset{\oplus}{+} bx^2$ defined over $\mathbb{Z}_{2^n}$ is self-inverse if the following conditions are satisfied:*

1. $a = 01\underbrace{\cdots 1}_{n-1}$ *or* $a = 1\underbrace{1\cdots 1}_{n}$ *in* $\mathbb{Z}_{2^n}$

2. *For n even, $b = \begin{smallmatrix}1\\0\end{smallmatrix} \cdots \begin{smallmatrix}1\\0\end{smallmatrix} \underbrace{0\cdots 0}_{r}$ in $\mathbb{Z}_{2^n}$ where $r \geq \frac{n}{2}$*

3. *For n odd, $b = \begin{smallmatrix}1\\0\end{smallmatrix} \cdots \begin{smallmatrix}1\\0\end{smallmatrix} \underbrace{0\cdots 0}_{r}$ in $\mathbb{Z}_{2^n}$ where $r \geq \frac{n-1}{2}$.*

**Proof.** First, it is required to prove that: If n is even, then $P(x) = ax + bx^2$ is a SIQPF over $\mathbb{Z}_{2^n}$, where $a = 01\underbrace{\cdots 1}_{n-1}$. Other cases can be proved in a similar way.

Let,

$$P(P(x)) = P(ax + bx^2)$$

and,

$$P(P(x)) = a \cdot (ax + bx^2) + b \cdot (ax + bx^2)^2$$

so,

$$P(P(x)) = a \cdot ax + a \cdot bx^2 + b \cdot a^2x^2 + 2ab^2x^3 + b^3x^4$$

or,

$$P(P(x)) = a^2x + (ab + a^2b)x^2 + 2ab^2x^3 + b^3x^4$$

Applying mod $2^n$ on both sides results in,

$$P(P(x)) \bmod 2^n = (a^2x + ab(1+a)x^2 + 2ab^2x^3 + b^3x^4) \bmod 2^n$$

From Lemma 4, the first term: $(a^2x) \bmod 2^n = 1 \cdot x = x$;
the second term: $(ab(1+ab)x^2) = (ab \cdot 2^{n-1}x^2) \bmod 2^n = 0$, where $(n-1) + n/2 > n$;
the third term: $(2ab^2x^3) \bmod 2^n = 0$, where $2r \geq 2 \cdot \frac{n}{2} = n$; and

the fourth term: $(b^3x^4)\bmod 2^n = 0$, where $3r \geq 3 \cdot \frac{n}{2} > n$,
which implies

$$P(P(x))\bmod 2^n = x \text{ for any } n, \qquad \text{Q.E.D.}$$

□

The same theorem for the formula in Equation (8) can be correspondingly proven and stated as follows:

**Theorem 5.** *For $n > 2$ an integer. $P(x) = a \stackrel{\oplus}{+} bx \stackrel{\oplus}{+} cx^2$ is a self-inverse permutation function defined over $\mathbb{Z}_{2^n}$ if the following conditions are satisfied:*

1.  $b = 01 \cdots 1 \underbrace{}_{n-1} or\ b = 11 \cdots 1 \underbrace{}_{n}\ in\ \mathbb{Z}_{2^n}.$

2.  *For $n$ even, $a = \begin{smallmatrix}1\\0\end{smallmatrix} \cdots \begin{smallmatrix}1\\0\end{smallmatrix} \underbrace{0 \cdots 0}_{i} and\ c = \begin{smallmatrix}1\\0\end{smallmatrix} \cdots \begin{smallmatrix}1\\0\end{smallmatrix} \underbrace{0 \cdots 0}_{j} in\ \mathbb{Z}_{2^n} where\ i, j \geq \frac{n}{2}.$*

3.  *For $n$ odd, $a = \begin{smallmatrix}1\\0\end{smallmatrix} \cdots \begin{smallmatrix}1\\0\end{smallmatrix} \underbrace{0 \cdots 0}_{i} and\ c = \begin{smallmatrix}1\\0\end{smallmatrix} \cdots \begin{smallmatrix}1\\0\end{smallmatrix} \underbrace{0 \cdots 0}_{j} in\ \mathbb{Z}_{2^n} where\ i, j \geq \frac{n-1}{2}.$*

**Proof.** First, it is required to prove that: If $n$ is even, then $P(x) = a \stackrel{\oplus}{+} bx \stackrel{\oplus}{+} cx^2$ is a self-inverse permutation function over $\mathbb{Z}_{2^n}$, where $b = 01 \cdots 1 \underbrace{}_{n-1}$. Other cases can be proved in a similar fashion.

Let $g(x)$ be a function which is defined as,

$$g(x) = P(x) - a = bx + cx^2$$

From Theorem 4, $g(x)$ is SIQPF.
Let,

$$P(P(x)) = P(a + bx + cx^2)$$

And

$$P(P(x)) = a + b \cdot (a + bx + cx^2) + c \cdot (a + bx + cx^2)^2$$

Yielding

$$P(P(x)) = a + ab + b \cdot (bx + cx^2) + a^2c + 2ac(bx + cx^2) + c \cdot (a + bx + cx^2)^2$$

By further simplification,

$$P(P(x)) = a(1 + b + ac) + 2ac(bx + cx^2) + g(g(x))$$

Applying mod $2^n$ on both and from Lemma 4 sides results in:
The first term $a(1 + b + ac)\bmod 2^n = (a \cdot 2^{n-1} + a^2c)\bmod 2^n = 0$, where $i + (n-1) \geq n/2 + (n-1) > n$ and $2i + j \geq n + n/2 > n$,
The second term: $2ac(bx + cx^2)\bmod 2^n = 0$, where $a \cdot c = 0$ for $i + j \geq n/2 + n/2 = n$.
And the third term: $g(g(x))\bmod 2^n = x$.
That implies, $P(P(x))\bmod 2^n = x$ for any integer $n$, □ Q.E.D.

Unfortunately, the $P(x)$ class of Theorem 4 suffers from the same fixed-points weakness as that in $P(x)$ of Theorem 2. In other words, there exist two specific fixed points ($x = 0, 2^{n-1}$) for every SIQPF. A remedy for that weakness is proposed in this work to remove the specific constant fixed points in

Theorem 4. This is attained similarly as in [55] by deploying the Boolean operator (OR) as proved in the following theorem.

**Theorem 6.** *For n > 2. A polynomial $P(x) = ax \overset{\oplus}{+} b(x^2 \vee D)$ is a SIQPF in $\mathbb{Z}_{2^n}$ if it satisfies all the following conditions:*

1.  $a = 01 \underbrace{\cdots 1}_{n-1}$ *or* $a = 11 \underbrace{\cdots 1}_{n}$ *in $\mathbb{Z}_{2^n}$.*

2.  *For n even,* $b = \begin{smallmatrix} 1 \\ 0 \end{smallmatrix} \cdots \begin{smallmatrix} 1 \\ 0 \end{smallmatrix} \underbrace{0 \cdots 0}_{r}$ *in $\mathbb{Z}_{2^n}$ where $r \geq \frac{n}{2}$.*

3.  *For n odd,* $b = \begin{smallmatrix} 1 \\ 0 \end{smallmatrix} \cdots \begin{smallmatrix} 1 \\ 0 \end{smallmatrix} \underbrace{0 \cdots 0}_{r}$ *in $\mathbb{Z}_{2^n}$ where $r \geq \frac{n-1}{2}$.*

4.  *D is an integer number.*

*Moreover, the resulting two fixed points of any P(x) are distinct and different for each individual SIQPF.*

**Proof.** Let us prove that $P(x) = ax \overset{\oplus}{+} b(x^2 \vee D)$ is SIQPF where $n$ is an even number and $a = 11 \underbrace{\cdots 1}_{n}$ in $\mathbb{Z}_{2^n}$. (Other cases can be proven in a similar way).

According to Theorems 4 and 5, the function $f(x) = ax + b \cdot g(x)$ is a SIQPF, where $g(x) = x^2$. This is always true for $g(x) = x^2 \vee D$, if $f(x) = ax + b \cdot g(x)$ is still an invertible function. And it is very easy to check if $f(x) = ax + b \cdot g(x)$ is an invertible function by using a bit-slice method (Lemma 2),

$$[f(x)]_0 = [ax + b \cdot g(x)]_0 = [ax]_0 \oplus [b \cdot g(x)]_0,$$

and

$$[f(x)]_0 = ([a]_0 \wedge [x]_0) \oplus ([b]_0 \wedge [g(x)]_0) = [x]_0, \tag{12}$$

where, $[a]_0 = 1$, and $[b]_0 = 0$. For $i > 0$,

$$[f(x)]_i = [ax + b \cdot g(x)]_i = [ax]_i \oplus [b \cdot g(x)]_i,$$

and

$$[f(x)]_i = ([a]_i \alpha_{[x]_0} \oplus [x]_i \alpha_{[a]_0} \oplus \alpha_{a \cdot x}) \oplus ([b]_i \alpha_{[g(x)]_0} \oplus [g(x)]_i \alpha_{[b]_0} \oplus \alpha_{b \cdot g(x)}).$$

First, we check the case of $0 < i < n/2$:

$$[f(x)]_i = (\alpha_{[x]_0} \oplus [x]_i \alpha_{[a]_0} \oplus \alpha_{a \cdot x}) \oplus (\oplus \alpha_{b \cdot g(x)}),$$

and

$$[f(x)]_i = [x]_i \oplus \beta, \tag{13}$$

where, $\beta = \alpha_{[x]_0} \oplus \alpha_{a \cdot x} \oplus \alpha_{b \cdot g(x)}$. Now, for $i \geq n/2$,

$$[f(x)]_i = (\alpha_{[x]_0} \oplus [x]_i \alpha_{[a]_0} \oplus \alpha_{a \cdot x}) \oplus ([b]_i \alpha_{[g(x)]_0} \oplus \alpha_{b \cdot g(x)}),$$

And

$$[f(x)]_i = [x]_i \oplus \gamma, \tag{14}$$

where, $\gamma = \alpha_{[x]_0} \oplus \alpha_{a \cdot x} \oplus [b]_i \alpha_{[g(x)]_0} \oplus \alpha_{b \cdot g(x)}$.

Technically, from Equations (12)–(14) and Definition 3, the function $f(x) = ax + b \cdot g(x)$ can be represented as,

$$[f(x)]_i = [x]_i + \alpha,$$

which holds for any bit $i$ and $\alpha$ is a parameter. Therefore, $f(x) = ax + b \cdot (x^2 \vee D)$ is an invertible function.

It is very simple to show that $b((2^{n-1})^2 \vee D) = bD$ and $b((0)^2 \vee D) = bD$ over $\mathbb{Z}_{2^n}$, therefore $x = 0, 2^{n-1}$ are not fixed points for any $D \neq 0$. Suppose now, for $i = 1, 2$, there are two different SIQPFs such as $P_i(x) = ax + b(x^2 \vee D_i)$, where $D_1 \neq D_2 \neq 0$, having the same fixed point $x_0$ as follows:

$$P_1(x_0) = P_2(x_0),$$

which implies,

$$x_0^2 \vee D_1 = x_0^2 \vee D_2.$$

The last step is only correct, if and only if $D_1 = D_2$, which proves that the resulting fixed points of any P(x) are distinct and different for each individual SIQPF, Q.E.D. □

*Practical significance of the SIQPFs of Theorem 6*

The fact that the resulting SIQPF according to the construction in Theorem 6, results with two individual and different fixed points for each different SIQPF is advantageous for cryptographic applications. The reason is that, ciphering operations involve usually cascading many different SIQPFs as round functions (see cipher structure in Section 9 and Figure 8). Therefore, the dynamic distribution of the different fixed points for different SIQPFs in different cascading stages results in general with improved random diffusion property of the overall cipher permutation.

Extending the $P(x)$ class of Equation (9) similarly as done for the class of Equation (8) in Theorem 6, results in a new larger class with similar properties as in the following Theorem 7:

**Theorem 7** *For n >2 an integer.* $P(x) = a \overset{\oplus}{\underset{+}{}} bx \overset{\oplus}{\underset{+}{}} c(x^2 \vee D)$ *is a self-inverse permutation function defined over* $\mathbb{Z}_{2^n}$ *if the following conditions are satisfied:*

1.   $b = 01 \cdots 1$ *or* $b = 11 \cdots 1$ *in* $\mathbb{Z}_{2^n}$.
     $\underbrace{\phantom{01\cdots1}}_{n-1} \quad \underbrace{\phantom{11\cdots1}}_{n}$

2.   *For n even,* $a = \begin{smallmatrix}1\\0\end{smallmatrix} \cdots \begin{smallmatrix}1\\0\end{smallmatrix} \underbrace{0 \cdots 0}_{i}$ *and* $c = \begin{smallmatrix}1\\0\end{smallmatrix} \cdots \begin{smallmatrix}1\\0\end{smallmatrix} \underbrace{0 \cdots 0}_{j}$ *in* $\mathbb{Z}_{2^n}$ *where* $i, j \geq \frac{n}{2}$.

3.   *For n odd,* $a = \begin{smallmatrix}1\\0\end{smallmatrix} \cdots \begin{smallmatrix}1\\0\end{smallmatrix} \underbrace{0 \cdots 0}_{i}$ *and* $c = \begin{smallmatrix}1\\0\end{smallmatrix} \cdots \begin{smallmatrix}1\\0\end{smallmatrix} \underbrace{0 \cdots 0}_{j}$ *in* $\mathbb{Z}_{2^n}$ *where* $i, j \geq \frac{n-1}{2}$.

4.   *D is an integer number.*

     *Moreover, the resulting two fixed points of any P(x) are distinct and different for each individual SIQPF.*

**Proof.** Is similar to that of Theorem 6. □

## 7. Cardinality of Proposed SIQPF Classes

In this section, the cardinality of the SIQPF classes are evaluated. Moreover, the equivalent and distinct mappings of the permutation polynomials are identified.

In $\mathbb{Z}_{2^n}$, not all permutations can be generated by polynomials and every permutation may be generated by different polynomials which are called equivalent polynomials modulo $2^n$. Therefore, computing the number of distinct polynomial permutations over $\mathbb{Z}_{2^n}$, requires excluding equivalent cases.

Let $P_n$ be a set of all possible permutation polynomials resulting with distinct permutations over $\mathbb{Z}_{2^n}$. Keller et al. [57] presented a formula to determine the cardinality of $P_n$. The cardinality of the set of all polynomial functions over different rings with some special conditions is presented in [58]. The formula which determines the cardinality of $P_n$ is given in [57,59] as follows:

$$|P_n| = 2^{3 + \sum\limits_{k=3}^{n} \beta(k)},$$ (15)

where, $\beta(k)$ is the smallest integer s such that $2^k$ divides s!.

Making use of Rivest Theorem 1, the number of permutation polynomials of degree at most $d$ can be computed by the following lemma:

**Lemma 5.** *For n >2, the number of all possible permutation polynomials of degree at most d over $\mathbb{Z}_{2^n}$ is $N_0$ where:*

$$N_0 = 2^{n + d \cdot (n-1)}$$ (16)

**Proof.** For every $n$, and from Theorem 1, the following is always true:

$$a_0 = \underbrace{\begin{matrix} 1 & & 1 \\ 0 & \cdots & 0 \end{matrix}}_{n} \quad \Rightarrow \quad |a_0| = 2^n,$$

and

$$a_1 = \underbrace{\begin{matrix} 1 & & 1 \\ 0 & \cdots & 0 \end{matrix}}_{n-1} 1 \quad \Rightarrow \quad |a_1| = 2^{n-1}.$$

Using the fact, that the sum of two even or odd integers is always even and according to Theorem 1, implies:

$$a_i = \underbrace{\begin{matrix} 1 & & 1 \\ 0 & \cdots & 0 \end{matrix}}_{n-1} * \quad \Rightarrow \quad |a_i| = 2^{n-1},$$

where the * position is a fixed 0 or 1. That is, the number of permutation polynomials which have the form $P(x) = a_0 + a_1 x + \cdots + a_d x^d$ is:

$$N_0 = 2^n \cdot \underbrace{2^{n-1} \cdots 2^{n-1}}_{d} = 2^{n + d(n-1)}, \text{ Q.E.D.}$$

□

**Definition 6.** *The polynomials f(x) and g(x) over $\mathbb{Z}_{2^n}$ are equivalent polynomials modulo $2^n$, if such polynomials satisfy the following condition [60]:*

$$f(x) \equiv g(x) \bmod 2^n$$ (17)

*In other words, the resulting polynomials generates the same permutation. Note that according to the above lemma, the number of permutation polynomials $N_0$ may include some equivalent permutation polynomials modulo $2^n$. The following definition appears to be useful for the targeted evaluation.*

**Definition 7.** *The cardinality of the set of all equivalent permutation polynomials modulo $2^n$ with degree $d \le 2^n - 1$, is equal to the numbers of all possible permutation polynomials $N_0$ having the degree $d \le 2^n$-1 excluding all distinct permutation polynomials $|P_n|$. That is:*

$$\left| Equivalent\ Permutations \right| = N_0 - \left| P_n \right|$$ (18)

*Table 1 shows the number of equivalent permutation polynomials modulo $2^n$ of degree at most $2^n - 1$ for few selected small values of n. It is noticed that for even small n = 8 results with a huge number of equivalent permutation polynomials.*

**Table 1.** Number of distinct and equivalent permutation polynomials (PPs).

| $\mathbb{Z}_{2^n}$ | # of Distinct PPs (15) | # of Equivalent PPs $N_0$ - $|P_n|$ |
|---|---|---|
| $n = 4$ | $2^{13}$ | $2^{48.8}$ |
| $n = 6$ | $2^{29}$ | $2^{321}$ |
| $n = 8$ | $2^{47}$ | $2^{17923}$ |

Therefore, it seems useful to seek an upper bound for the degree $d$ of all distinct permutation polynomials. The following upper bound on $d$ can be derived by making use of (16) for ($N_0 = |P_n|$), resulting with:

$$\widetilde{d} = \left\lceil \frac{\log_2 |P_n| - n}{n - 1} \right\rceil, \tag{19}$$

where the degree $\widetilde{d}$ is the upper bound of the degree of distinct PPs in size $n$. The values of $n$, $P_n$ are known and $\lceil \cdot \rceil$ is the ceiling function. The formula in Equation (19) represents a necessary design rule for selecting such distinct permutation polynomials.

Table 2 shows the relation between the cardinality of PPs and the corresponding highest degree d of non-equivalent PPs over $\mathbb{Z}_{2^n}$. As all practical applications require n > 3 for $\mathbb{Z}_{2^n}$, and all proposed new SIQPFs classes have degree 2 then all resulting PPs in any class are distinct. In the following, the cardinality of each distinct class of the new SIQPFs is computed:

**Table 2.** Upper bound of degree of distinct PPs.

| $\mathbb{Z}_{2^n}$ | Maximum Degree $\widetilde{d}$ |
|---|---|
| $n = 4$ | 3 |
| $n = 5$ | 4 |
| $n = 6$ | 5 |
| $n = 7$ | 5 |
| $n = 8$ | 6 |

**Corollary 1.** *For n > 2, the cardinality of a class C1:$P(x) = ax \overset{\oplus}{+} bx^2$ over $\mathbb{Z}_{2^n}$ is:*

$$|C_1| = \begin{cases} 2(2^{\frac{n}{2}} - 1) & : & for\, n\ even \\ 2(2^{\frac{n+1}{2}} - 1) & : & fon\, n\ odd \end{cases} \tag{20}$$

**Proof.** For $n$ even, and from Theorem 4, the following is true:

$$a = 0\underbrace{1\cdots 1}_{n-1}\ or\ \underbrace{1\cdots 1}_{n} \quad \Rightarrow \quad |a| = 2,$$

and

$$b = \begin{matrix} 1 \\ 0 \end{matrix} \cdots \begin{matrix} 1 \\ 0 \end{matrix} \underbrace{0\cdots 0}_{\frac{n}{2}} \quad \Rightarrow \quad |b| = 2^{\frac{n}{2}-1}.$$

This implies, $|C_1| = |a| \cdot |b| = 2(2^{\frac{n}{2}} - 1)$.

For $n$ odd and applying similar steps, that implies:$|C_1| = |a| \cdot |b| = 2(2^{\frac{n+1}{2}} - 1)$,    Q.E.D.   □

Table 3 shows the corresponding cardinalities for all new SIQPFs classes, where the procedure of Corollary 1 is repeatedly applied for each class. For a practical example of 64-bit arithmetic where $n = 64$, the cardinalities of the permutation classes $|C_1| = 2^{33}$, $|C_2| = 2^{65}$, $|C_3| = 2^{97}$ and $|C_4| = 2^{129}$.

**Table 3.** Cardinality of all resulting classes of Self-Inverting Permutation Functions (SIPFs).

| Class of SIPFs in $\mathbb{Z}_{2^n}$ | Cardinality | |
| --- | --- | --- |
| | *n*: Even | *n*: Odd |
| $C_1{:}P(x) = ax \overset{\oplus}{+} bx^2$ | $\geq 2(2^{\frac{n}{2}} - 1)$ | $\geq 2(2^{\frac{n+1}{2}} - 1)$ |
| $C_2{:}P(x) = a \overset{\oplus}{+} bx \overset{\oplus}{+} cx^2$ | $\geq 2(2^{\frac{n}{2}} - 1)^2$ | $\geq 2(2^{\frac{n+1}{2}} - 1)^2$ |
| $C_3{:}P(x) = ax \overset{\oplus}{+} b(x^2 \vee D)$ | $\geq 2^{n+1}(2^{\frac{n}{2}} - 1)$ | $\geq 2^{n+1}(2^{\frac{n+1}{2}} - 1)$ |
| $C_4{:}P(x) = a \overset{\oplus}{+} bx \overset{\oplus}{+} c(x^2 \vee D)$ | $\geq 2^{n+1}(2^{\frac{n}{2}} - 1)^2$ | $\geq 2^{n+1}(2^{\frac{n+1}{2}} - 1)^2$ |

Notice that the given cardinalities in Table 3 represent worst case bounds. The exact cardinalities seem to be difficult to evaluate as equal mapping may happen in different mappings constellations. Therefore, the smallest cardinality values are used when evaluating the resulting cipher performance and cardinalities.

## 8. Hardware and Complexity Evaluation of SUC Rounds

To implement SQIPFs in the targeted FPGA platform, fabric Look-Up Tables (LUTs), D-Flip Flops (DFFs), and Mathblocks are required. Therefore, an optimal and effective implementation strategy proposes to attain the same number of LUTs and DFFs, i.e. the ratio $R_{LUT/DFF} = \frac{\#of\ LUTs}{\#of\ DFFs}$ should be close to 1. This can be inferred from the fact that when consuming an LUT, its corresponding DFF in the same logic cell cannot be used elsewhere as its input is used for the LUT. Most FPGA architectures provide an easy to connect DFF within each LUT.

The implementation complexity of such classes in SmartFusion®2 SoC FPGA is one of the major objectives of this research. Therefore, a sample complexity evaluation should show the relative efficiency of the designed permutations.

These classes of SIQPFs could be implemented in both hardware and software or by combining HW/SW implementation scenario for the targeted SUCs described in [29]. In this case, the SIQPF could constitute one efficient class required for constructing the SUC cascade. The cryptographic strength (attack complexity) of the generated permutations is attained through a huge number of possibilities of each SIQPF class simply controlled by the permutation function coefficients.

### 8.1. Hardware Complexity

Figure 7 shows a basic hardware configuration for building the function $a + bx + cx^2$ for $n = 18$ bits by using 2 Mathblocks. The designed SIQPFs are modeled in VHDL and synthesized to check their hardware complexity and performance. ModelSim ME package is used for simulation and Synplify pro ME within Libero SoC is used for synthesis. When implementing these functions, the Multipliers (MACC), LUT and DFF constitute the basic resources for implementing such functions.

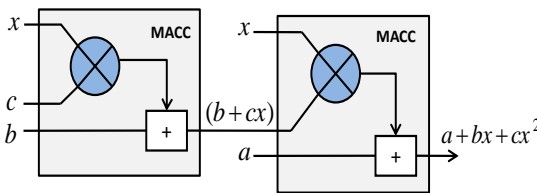

**Figure 7.** Implementing $a + bx + cx^2$ by using two Mathblocks.



By analyzing the FPGA resource usage for each function, a closed formula was found for the number of MACCs, ($N_{MACC}$), LUTs ($N_{4LUT}$) and DFFs ($N_{DFF}$) for each class of SIQPFs and input data size in bits. Table 4 shows the required hardware resources (hardware complexity) as a function of the number of bits $n$ for $1 \leq n \leq 32$ bits for the permutation classes $C_1$ and $C_2$. The complexities of the permutation classes $C_3$ and $C_3$ are slightly higher and are not included in this evaluation.

**Table 4.** Hardware complexity. LUT = Look-Up Tables; DFF = D-Flip Flops; $N_{MACC}$ = number of MACCs; $N_{4LUT}$ = number of Look-Up Tables; $N_{DFF}$ = number of D-Flip Flops.

| | |
|---|---|
| $a + bx + cx^2$ | $N_{MACC} = 2U(n-3) + U(n-9) + 6U(n-18)$<br>$N_{4LUT} \quad = U(n-1) + 3U(n-2) + 69U(n-3)$<br>$\qquad +36U(n-9) \; + (2n+217)U(n-18)$ |
| | $N_{DFF} = 72U(n-3) + 36U(n-9) + 216U(n-18)$ |
| $a + bx - cx^2$ | $N_{MACC} = 3U(n-3) + 6U(n-18)$ |
| | $N_{4LUT} \quad = U(n-1) + 3U(n-2) + 109U(n-3)$<br>$\qquad +4U(n-4) \; + (2n+217)U(n-18)$ |
| | $N_{DFF} = 108U(n-3) + 216U(n-18)$ |
| $bx + cx^2$ | $N_{MACC} = 2U(n-3) + U(n-9) + 6U(n-18)$ |
| | $N_{4LUT} \quad = U(n-1) + 2U(n-2) + 70U(n-3)$<br>$\qquad +36U(n-9) \; + (n+217)U(n-18)$ |
| | $N_{DFF} = 72U(n-3) + 36U(n-9) + 216U(n-18)$ |
| $bx - cx^2$ | $N_{MACC} = 3U(n-3) + U(n-9) + 9U(n-18)$ |
| | $N_{4LUT} \quad = U(n-1) + 2U(n-2) + 109U(n-3)$<br>$\qquad -3U(n-4) + (n+217)U(n-18)$ |
| | $N_{DFF} = 108U(n-3) + 216U(n-18)$ |
| $a + bx \oplus cx^2$ | $N_{MACC} = 3U(n-3) + 9U(n-18)$ |
| | $N_{4LUT} \quad = U(n-1) + 2U(n-2) + (n+109)U(n-3)$<br>$\qquad +(n+199)U(n-18)$ |
| | $N_{DFF} = 108U(n-3) + 216U(n-18)$ |
| $bx \oplus cx^2$ | $N_{MACC} = 3U(n-3) + 9U(n-18)$ |
| | $N_{4LUT} \quad = U(n-1) + 3U(n-2) + (n+105)U(n-3)$<br>$\qquad +(n+216)U(n-18)$ |
| | $N_{DFF} = 108U(n-3) + 216U(n-18)$ |

Here $U$ is a unit step function defined as $U(t - t_0) = \begin{cases} 0 : t < t_0 \\ 1 : t \geq t_0 \end{cases}$

Table 5 shows sample hardware complexities for permutations with number of bits $n$ up to 32 bits. Notice, $R_{LUT/DFF}$ of class ($bx - cx^2$), for instance, is equal to 1.01, 1.009, and 1.1, when $n$ =8, 17 and 32 respectively.

Notice also that, SIQPF $a + bx + cx^2$ exhibits the highest efficiency as it makes maximum use out of the same deployed number of MACCs.

**Table 5.** Sample hardware complexity evaluations.

| SIPFs | | Total Cost | | | |
|---|---|---|---|---|---|
| | | $n = 8$ | $n = 17$ | $n = 32$ | $R_{LUT/DFF}$, $n = 32$ |
| $a + bx + cx^2$ | $N_{MACC}$ | 2 | 3 | 9 | 1.2 |
| | $N_{4LUT}$ | 73 | 109 | 390 | |
| | $N_{DFF}$ | 72 | 108 | 324 | |
| $a + bx - cx^2$ | $N_{MACC}$ | 3 | 3 | 9 | 1.2 |
| | $N_{4LUT}$ | 109 | 109 | 390 | |
| | $N_{DFF}$ | 108 | 108 | 324 | |
| $bx + cx^2$ | $N_{MACC}$ | 2 | 3 | 9 | 1.1 |
| | $N_{4LUT}$ | 73 | 109 | 357 | |
| | $N_{DFF}$ | 72 | 108 | 324 | |
| $bx - cx^2$ | $N_{MACC}$ | 3 | 3 | 9 | 1.1 |
| | $N_{4LUT}$ | 73 | 109 | 358 | |
| | $N_{DFF}$ | 72 | 108 | 324 | |
| $a + bx \oplus cx^2$ | $N_{MACC}$ | 3 | 3 | 9 | 1.1 |
| | $N_{4LUT}$ | 117 | 126 | 357 | |
| | $N_{DFF}$ | 108 | 108 | 324 | |
| $bx \oplus cx^2$ | $N_{MACC}$ | 3 | 3 | 9 | 1.2 |
| | $N_{4LUT}$ | 117 | 126 | 389 | |
| | $N_{DFF}$ | 108 | 108 | 324 | |

## 8.2. Software Complexity

SmartFusion®2 SoC FPGA incorporates ARM Cortex-M3 that supports Thumb2 instruction set, it contains enhanced instructions as single cycle multiplication between two numbers of 32 bits. Table 6 shows the time and memory implementation complexities of the same set of permutation functions when using ARM Cortex software environment for some chosen numbers of bits $n$.

**Table 6.** Software performance and complexity.

| SIPFs | Total Cost | |
|---|---|---|
| | $n = 16$ | $n = 32$ |
| $bx + cx^2$ | 36 bytes<br>15 cycles | 28 bytes<br>13 cycles |
| $bx - cx^2$ | 36 bytes<br>15 cycles | 28 bytes<br>13 cycles |
| $bx \oplus cx^2$ | 84 bytes<br>29 cycles | 36 bytes<br>16 cycles |
| $a + bx + cx^2$ | 60 bytes<br>22 cycles | 36 bytes<br>18 cycles |
| $a + bx - cx^2$ | 76 bytes<br>27 cycles | 36 bytes<br>18 cycles |
| $a + bx \oplus cx^2$ | 100 bytes<br>34 cycles | 44 bytes<br>19 cycles |

## 9. Proposed New SUC Constructions Based on Self-Inverse Permutations

To generate larger classes of ciphers, cascading one or more SIQPFs is necessary. This is also the first step toward creating SUCs with cryptographically significant entropy.

### 9.1. Possible Creation of an SUC as a Key-Alternating Cascade of SIQPFs

In [56], the permutation T-function $f$ used to construct unusual permutations by Xoring of any pair of $f(x)$, $x$ as $f(x) \oplus x$. Furthermore, extending the resulting SIQPF's classes by using this result, for example, with $P(x) = a + bx \oplus cx^2$ as a SIQPF, the XOR of any pairs of P(x), $x$ is a permutation $P(x) \oplus x$ but not necessarily a SIQPF. One of the most important results based on this discussion is in deploying them in counteracting the weakness stated in Theorems 4–7. For example, let $P(x) = 3x + 2x^2$ be a SIQPF in $\mathbb{Z}_{2^3}$ and let $G(x) = x \oplus 3$ be an XOR mapping where the bitwise XORing operation with any given value represents an involution. Therefore, the resulting function composition $f(x) = (3x + 2x^2) \oplus 3$ is SIQPF without fixed points. However, the previous discussion proves that XORing a SIQPF with the key k results with in a round function $P(x) \oplus k$ for a block cipher which avoids fixed points, where $k$ is a round key.

#### 9.1.1. Cardinality of the GENIE-Selectable SUCs

This section investigates the cardinality of the resulting cipher when using the key-alternating cipher structure of Figure 8. The cipher has $r$ randomly selected self-inverse permutations $P_1, \dots, P_r$ and $(r + 1)$ randomly selected $n$-bit keys.

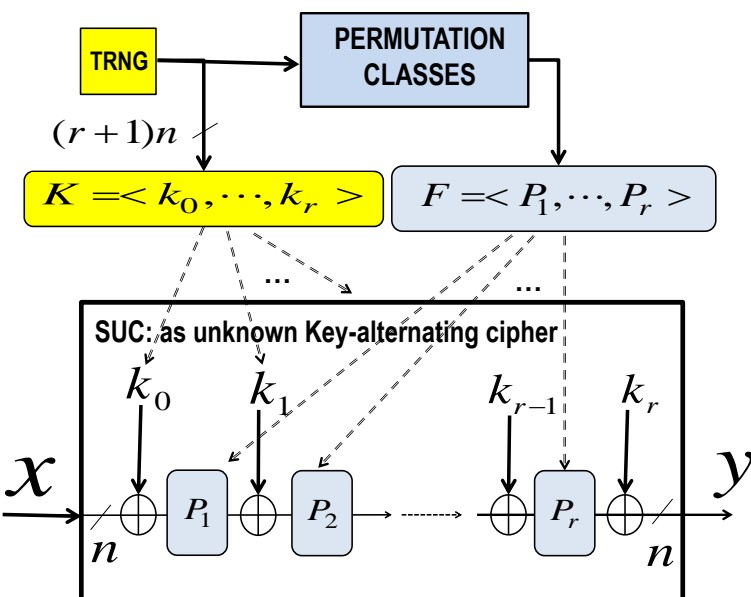

**Figure 8.** Key-alternating format of the targeted GENIE-created SUCs.

The usable self-inverse permutations are included in the above 4 different classes $C_1$, $C_2$, $C_3$, and $C_4$ from which $P_1, \dots, P_r$ can be randomly selected.

Notice that, for highest security and to avoid fixed points in the total mapping, at least one permutation needs to be selected from the classes $C_3$ *and/or* $C_4$. We select few possible random cipher selection strategies to evaluate the cardinality of all possible selectable SUCs.

A.　Fully selecting from the set containing all 4 classes

In this case, we consider the selectable mappings, in Figure 8, to be out of the set of all 4 classes $C_1$, $C_2$, $C_3$, and $C_4$. Hence:

- The cardinality $S$ of all classes of mappings is then:

$$S = |C_1| + |C_2| + |C_3| + |C_4| = (2^{\frac{n+1}{2}} - 1)\left(2^{\frac{n+1}{2}+1} + 2^{\frac{(n+1)^2}{2}}\right)$$

- As each cipher utilizes $(r+1)$ keys of size $n$, the cardinality of keys is: $2^{n \times (r+1)}$
- The GENIE selects $r$-mappings randomly from S, hence there is $r!$ possible placements of the mappings to build each cipher.

The GENIE selects randomly $r$ mappings from the classes of mappings $S$ and $(r+1) \times n$ key bits. The placement of the selected mappings is totally random. We investigate the following two selection cases.

**Case 1.** *If the selection is done with allowed Pi repetition (i.e. the GENIE can select the same mapping multiple times for different Pi s), then the total number $\sigma_{11}$ of all possible different key-alternating ciphers-selections including keys according to Figure 8 as an SUC having r rounds with an input size of n-bit is:*

$$\sigma_{11} = r! \times S^r + 2^{n \times (r+1)}$$

**Case 2.** *If the selection is done without Pi repetition, then the total SUC selections cardinality $\sigma_{12}$ in that case according to Figure 8 is:*

$$\sigma_{12} = r! \times \binom{S}{r} + 2^{n \times (r+1)}$$

B.    Selecting fixed number of mappings from each class

In the second case, we consider for lower hardware complexity that the GENIE is designing the SUC cipher by deploying a fixed number of selections for $P_i$ from each class $C_i$, the realization of such SUCs is according to the SUC-Design-Template mechanism proposed in [30]. Since there exist 4 classes, we consider that the cipher-structure has a fixed template of $P_i$ selections as follows:

- $t_1$ mappings ($P_i$ s) from $C_1$
- $t_2$ mappings ($P_i$ s) from $C_2$
- $t_3$ mappings ($P_i$ s) from $C_3$
- $t_4$ mappings ($P_i$ s) from $C_4$

Note that $t_1 + t_2 + t_3 + t_4 = r$. $t_i \geq 0$

**Case 3.** *If the selection from each class $C_i$, with $1 \leq i \leq 4$, is done with repetition, then the total number $\sigma_{21}$ of all possible different key-alternating ciphers-selections according to Figure 8 as an SUC having r rounds with an input size of n bit is:*

$$\sigma_{21} = (t_1! \times t_2! \times t_3! \times t_4!) \times \left(\left|C_1\right|_{t_1} \times \left|C_2\right|_{t_2} \times \left|C_3\right|_{t_3} \times \left|C_4\right|^{t_4}\right) + 2^{n \times (r+1)}$$

$$\sigma_{21} = \prod_{i=1}^{4}\left(t_i! \times |C_i|^{t_i}\right) + 2^{n \times (r+1)}$$

**Case 4.** *If the selection is done without repetition, then the cardinality $\sigma_{22}$ of this cipher (Figure 8) is:*

$$\sigma_{22} = (t_1! \times t_2! \times t_3! \times t_4!) \times \binom{|C_1|}{t_1} \times \binom{|C_2|}{t_2} \times \binom{|C_3|}{t_3} \times \binom{|C_4|}{t_4} + 2^{n \times (r+1)}$$

$$\sigma_{22} = \prod_{i=1}^{4} \left( t_i! \times \binom{|C_i|}{t_i} \right) + 2^{n \times (r+1)}$$

The main advantage of deployed involutive (self-inverse) cascaded permutations (as SIQPFs) is that the same mapping modules can be used for both encryption and decryption operations by just reversing the sequence of the mappings with their round keys.

9.1.2. SUC Cloning Security Measure

SUC cloning security depends on the attacker's effort required to find the unknown selected SUC out of all possible choices $\sigma_{ij}$. For example, if the GENIE is selecting according to Case 1 creating a cipher with a block size n = 64 bit having r = 16 rounds, then:

$$\sigma_{11} = r! \times S^r + 2^{n \times (r+1)} = r! \times \left[ (2^{\frac{n+1}{2}} - 1)\left(2^{\frac{n+1}{2}+1} + 2^{\frac{(n+1)^2}{2}}\right) \right]^r + 2^{n \times (r+1)}$$

$$\sigma_{11} = 16! \times \left[ (2^{\frac{64+1}{2}} - 1)\left(2^{\frac{64+1}{2}+1} + 2^{\frac{(64+1)^2}{2}}\right) \right]^{16} + 2^{64 \times (16+1)} \approx \underbrace{2^{34904}}_{\text{Cipher cardinality}} + \underbrace{2^{1088}}_{\text{Key cardinality}}$$

That is the attacker needs to reveal the correct cipher-choice out of about $2^{34904}$ possible SUC-choices (without considering the key-choices) to be able to clone it. Notice that a part of the resulting ciphers in different cascade constellation may result with equal ciphers. That is, the cipher cardinality may be reduced to become $2^{34904-d}$, where $d$ is hard to find. However $d$ is expected to be cryptographically not significant.

Therefore, for a larger $\sigma_{ij}$ more cloning security is attained. Notice that a larger $\sigma_{ij}$ mostly requires more hardware complexity.

*9.2. Security Evaluation of the Resulting SUCs*

The security level (or bound) of any cipher is conventionally determined by applying Kerckhoffs's principle. That is the attacker knows all details of the used cipher structure except the cipher-key which is unknown to the attacker. As in SUC concept, the cipher is even not known to anybody, the attack complexity is basically expected to increased [61]. However, the cipher structure may be predicted if the GENIE is published. Otherwise, if the GENIE is not published (this is allowed in the proposed realization concept), the attack complexity is expected to increase.

A well-known interpolation attack would successfully reveal a mapping y equivalent to the whole SIQPF cascade if and only if SUC is given as a cascade of only algebraic classes of SIQPF such as $P(x) = a \pm bx \pm cx^2$. In this case, an adversary can compute such equivalent mapping y to all r rounds by just 2r+1 known plaintext/ciphertext pairs [62,63] as follows:

$$y = f(x) = \sum_{i=1}^{r} \prod_{1 \leq j \leq r, i \neq j} \frac{x - x_j}{x_i - x_j} \tag{21}$$

Note that an adversary needs just to know that the functions as a quadratic one and guess the number of rounds *r*.

In another case, the interpolation attack is not applicable, when the selection of SIQPFs is drawn from non- algebraic classes such as $P(x) = a \overset{\oplus}{+} bx \overset{\oplus}{+} c(x^2 \vee D)$. However, Klimov et al. [55] presented an

attack scenario on T-function $f(x) = x + (x^2 \vee C)$, when it is as a substitute for LFSR in a stream cipher. This attack works, for example, only if the size of C is small such as $\frac{n}{3}$, and then it requires $O(2^{n/3})$. There is no such attack on a block cipher that uses a T-function as round function.

*9.3. Modeling Attack on the Proposed SUC*

When looking at modeling attacks on SUCs there are two possibilities: Firstly, the target of an adversary using ML is to create a predictive model of an SUC by analyzing some training data. Theoretically, if an SUC is a weak Pseudorandom Function (PRF), then certain patterns of plaintext/ciphertext pairs could be easily identified and detected by a ML algorithm with little training. But when an SUC is a secure PRF, the successful detection of patterns becomes impossible. Moreover, if a designed SUC is a secure PRF, then there is no ML algorithm that can build a predictive model for such an SUC, because the secure PRF concept postulates that the output of PRF is statistically independent of training data and uncorrelated with any learner [64].

The second possible modeling attack is to store all the possible plaintext/ciphertext pairs as the Cipher Codebook size CCBS=$2^n$. However, storing $2^n$ bits to build a model for an SUC is infeasible for ciphers with n > 80.

In this section, the focus is put on the adversary who tries to use the collected SUC-input/output pairs in distinguishing attacks. Here, successful distinguishing attacks on SUCs indicate that the designed SUC structure is vulnerable. Therefore, sooner or later the adversary can build a predictive model for the designed SUC. If not, modeling attacks on SUCs are almost infeasible.

As a result of the above notes, definitions, and discussion the self-generated SUC inside a chip can be modeled as a secure Pseudorandom Permutation (PRP) chosen randomly from the class of all possible generated ciphers $\{C_1, C_2 \ldots, C_\sigma\}$, where, $\sigma \leq 2^n!$ as follows:

$$SUC : \{0,1\}^n \times \{0,1\}^k \rightarrow \{0,1\}^n$$
$$(X,K) \overset{PRP}{\rightarrow} Y \tag{22}$$

where, n and k are the input-output size and the key size, respectively. The inverse of SUC should be a secure PRP as well:

$$SUC^{-1} : \{0,1\}^n \times \{0,1\}^k \rightarrow \{0,1\}^n$$
$$(Y,K) \overset{PRP}{\rightarrow} X \tag{23}$$

In [65], Bogdanov et al. conjectured that the query complexity of the distinguishing attack on a key-alternating cipher is $2^{\frac{r}{r+1}n}$, where, n is the cipher input size, and r is the rounds number. In [66], Steinberger presented an improved security bound $2^{\frac{3}{4}n}$ for the distinguishing attack on a key-alternating cipher, when $r > 2$. In [67], Lampe et al. showed that if r is even, the security bound attains $2^{\frac{r}{r+2}n}$. In [68], the tight security bound $2^{\frac{r}{r+1}n}$ of the distinguishing attack on a key-alternating cipher is proved, i.e., there is no more security bound that can be attained.

It is, therefore, conjectured, that the security bound $2^{(\varepsilon)n}$ is enhanced by a factor equal to the product of the cardinalities of the deployed SIQPFs, where $\varepsilon$ is a function of $r$. An ongoing research is conducted to answer this still open question. However, the proposed SUC cipher design fulfills, therefore, at least the state-of-the-art security requirements for standard good ciphers.

In the following section some statistical properties of SIQPFs are investigated. Here, the statistical properties provide an initial proof of the indistinguishability of the proposed SUCs.

*9.4. Statistical Properties of the Resulting SUC*

In this section, the diffusion [46] and a frequency prediction [69] as statistical properties of SIQPFs are analyzed and studied.

A.　　SUC Diffusion Properties

The essential definition of a diffusion is to determine the number of changed output bits when one input bit has been changed. Ideally, the changing ratio of the output bits is 50%. However, a T-function is defined as a mapping in which bit *i* of the output depends on 0, 1, ... , *i* bits of the inputs [54], thus indicating that changing the first input bit affects all n output bits, changing the second input bit affects the last n-1 output bits, etc. The changing of last input bit affects the last output bit. To test this property, the hamming distance between outputs of randomly selected SIQPF by changing one input bit every time. The applicable Algorithm 1 is defined as a simulator to determine the amount of diffusion.

---

**Algorithm 1** Diffusion Test on the Proposed SUCs.

---

Enter n = 32, cipher rounds r = 8, and select randomly one SIQPF P.
Select randomly 10000 input values $x(i)$, where $i = 1, \dots , 10000$.
Determine the dependence matrix A;
$$A_{ji}(P) = 10^{-4} \sum_{i=1}^{10000} P(x^{(i)}) \oplus P(x^{(i)} \oplus e_j),$$
where, $e_j = (\delta_{j1}, \delta_{j2}, \cdots, \delta_{j32})$, $\delta_{ji}$ is a Kronecker's delta.
Return the average of A.

---

The results in Figure 9 show that the increasing of number of rounds in Equation (21) does not change the statistical distribution of the diffusion of any resulting cipher in the class. In this case, the simulation indicates that the average of diffusion is close to 50% for repeatedly using a single SIPQF with r (iterations) rounds.

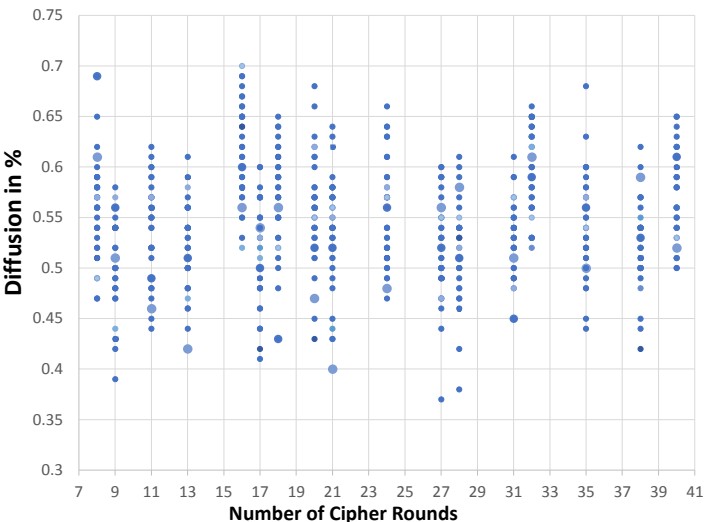

**Figure 9.** SIPQF classes diffusion by using 1000 random pairs.

B.    SUC Cipher Text Bit Frequency Distribution

A probability distribution $P[y_i = 1]$ indicates the level of predictability of the bit output $yi$, when $y_i$ =1. The ideal case corresponds to $P[y_i = 1] = 50\%$. To test this, 10000 random input/output pairs are used for selected SIPQF, where the predication of each output bit $y_i$ is given based on statistical distribution of $P[y_i = 1]$. This procedure has been performed and repeated for 10 randomly selected SIPQF. Results in Figure 10 show a high unpredictability of the bit output $y_i$, while $P[y_i = 1]$ is close to 50%.

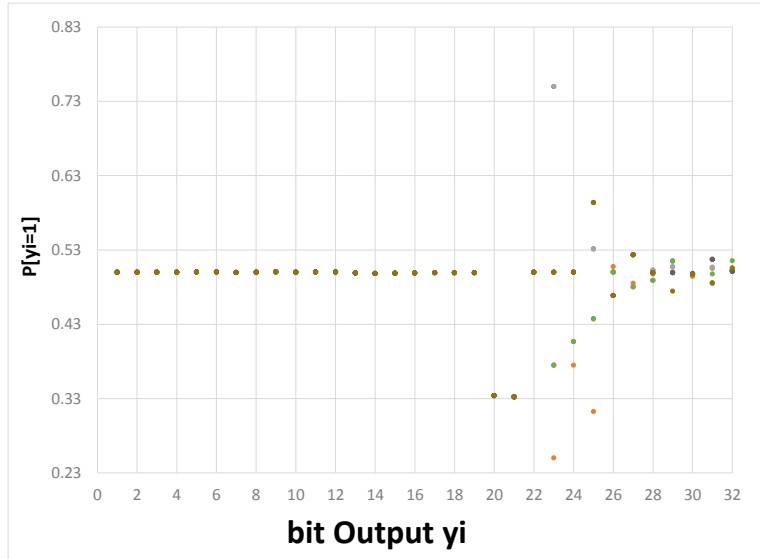

**Figure 10.** The probability that any output bit is 1.

## 10. Conclusions

In this paper, new classes of low-complexity self-inverse permutations based on T-Functions (SIPFs) are presented. The target usage of such functions is for creating the so-called Secret Unknown Ciphers (SUC) at very low cost to serve as clone resistant digital PUFs. The permutations algebra is deploying optimized usage of multiplier/Mathblocks units which are often not consumed in many FPGA applications. As a result, high quality ciphers may be embedded with negligible cost by using unconsumed resources in such SoC units. The resulting new cipher classes based on $\mathbb{Z}_{2^n}$ arithmetic are very promising in their security and quality. The resulting SUC structures are designed to be used as PUFs alternative and security anchors in the emerging future smart application scenarios. Creating SUCs is a very challenging task. This work is a part of an ongoing basic research towards creating and embedding SUCs in future VLSI-technologies and showing their possible applications. Being created in a manufacturer-independent-process and by end-users, the SUC technology is expected to be attractive for wide spectrum of applications in future automotive and IoT environment.

**Author Contributions:** Conceptualization, S.M., A.M and W.A.; Methodology, S.M. and A.M; Software, A.M and S.M.; Supervision, W.A.; Validation, S.M. and A.M.; Formal analysis, S.M. and A.M.; Visualization, S.M., A.M and W.A.; Writing—original draft preparation, S.M. and A.M; Writing—review and editing, S.M., W.A. and A.M.; Project administration, W.A. All authors have read and agreed to the published version of the manuscript.

**Funding:** This work was supported by the DAAD Research Grants-Doctoral Programmes in Germany Nr. (57214224) and the German Federal Foreign Office scholarship funding (STIBET) program, as well as Microsemi, a Microchip Company, San Jose, CA, USA, and Volkswagen AG-Germany.

**Conflicts of Interest:** The authors declare no conflict of interest.

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
