# Peer review of "Low-Complexity Nonlinear Self-Inverse Permutation for Creating Physically Clone-Resistant Identities"

_cryptography, doi:10.3390/cryptography4010006_

Round 1

Reviewer 1 Report

The intent of this paper is indeed interesting, and the notion of creating a PUF (or in your case a SUC) at the post manufacturing stage is clearly valuable. Doing so with the capabilities of a non-volatile FPGA based SoC is also intriguing. As I understand it, the intent of this paper is to introduce techniques to create the Secret Unknown Cipher within these devices in a cryptographically secure manner.

Regrettably, I am not expert in the mathematical formulations, and have little critique to offer there. But I have concerns about the writing presentation of the paper. There are many areas in which the language does not clearly convey the meaning, and at times, even distracts from your message. The cross disciplinary concepts you are presenting are rather obscure, and inherently challenging. As a result, the writing must be exceptional in order for the merit of your ideas to emerge. To meet this need, the paper needs a thorough editing alongside a native English speaker to ensure that those difficulties are solved.

Some Minor editing examples and suggestions.

Page 1 line 10 (abstract) – Perhaps change “special emerging near future” to “emerging”.

Page 1 line 27 – “early fallen short against”. Not sure the real intent here, but perhaps change to ”been shown vulnerable to”

Page 1 line 30 – “born” could be “native”

Page 1 Line 31 – “become equal” might be better as “be equivalent”

Page 1 Line 36 – delete “so far”

Page 2 Line 48 “PUD” should be “PUF”?

Page 2 Lines 75-80, The word “section” is used as a name here, so capitalize as “Section”

Page 2 Line 82, insert “it” between “as” and “is”, then end with period instead of comma.

Page 3 Line 112, cannot understand what it means to upload devices simultaneously with the Genie, need to explain better.

Page 3 Line 115, you mention True Random Generator (TRG), but do not explain or cite. “True” is a bold claim… needs support.

Page 3 Line 119, t1 and t2 seem to be superscript here, yet subscript elsewhere… be consistent.

Page 3 Line 126, discussion of sets of clear text and cipher text pairs, but no discussion on how big these sets must be

Page 5 Line 183, “technologies are expected to emerge”. As is, this whole sentence is unclear.

Page 3 Line 193 “tend to become an impossible mission”. Awkward at best. Perhaps “will be impossible”.

Page 5 Line 203, “. That is mainly not used” could be “; such as”

Page 6 Line 221, insert “able to“ before “irreversibly”.

(This list is by no means exhaustive, a thorough review is required.)

Reviewer 2 Report

The paper is well-written the ideas are clearly presented. The interest will increase towards the research when the current mostly theoretical work will have practical endorsements. 
I recommend the publication of the work after solving the followings:
1) The state-of-the-art in section 1 has to be augmented in the sense of a larger view and newer references.
It is mandatory (even if now the work relies more on the theoretical aspect) to make the connection of the presented ideas with the IIoT, the industry and other applications of PUF, respectively not to limit some issues to certain approaches (e.g. PUF with fuzzy).
I would consider at least the following supplementary references:
- Fladung, L.; Nikolopoulos, G.M.; Alber, G.; Fischlin, M. Intercept-Resend Emulation Attacks against a Continuous-Variable Quantum Authentication Protocol with Physical Unclonable Keys. Cryptography 2019, 3, 25.
- Calhoun, J.; Minwalla, C.; Helmich, C.; Saqib, F.; Che, W.; Plusquellic, J. Physical Unclonable Function (PUF)-Based e-Cash Transaction Protocol (PUF-Cash). Cryptography 2019, 3, 18.
- Zhu, F.; Li, P.; Xu, H.; Wang, R. A Lightweight RFID Mutual Authentication Protocol with PUF. Sensors 2019, 19, 2957.
- Chen, S.; Li, B.; Cao, Y. Intrinsic Physical Unclonable Function (PUF) Sensors in Commodity Devices. Sensors 2019, 19, 2428.
- Gołofit, K.; Wieczorek, P.Z. Chaos-Based Physical Unclonable Functions. Appl. Sci. 2019, 9, 991.
- Tidrea, A.; Korodi, A.; Silea, I. Cryptographic Considerations for Automation and SCADA Systems Using Trusted Platform Modules. Sensors 2019, 19, 4191.
- Furtak, J.; Zieliński, Z.; Chudzikiewicz, J. A Framework for Constructing a Secure Domain of Sensor Nodes. Sensors 2019, 19, 2797.
2) I would suggest to reformulate the sentences at lines 192-193 in a more academic manner:
"There is no doubt that both challenges are highly complex. However, there are no technical reasons to believe that SUC creation tend to become an impossible mission."

Round 2

Reviewer 1 Report

This revision demonstrates improvement in the presentation of your work. English structure at times is still lacking, and sometimes that interferes with understanding the work.

One general concern is the suggestion that after removing the Genie, only the SUC remains. Are you suggesting that the SUC is an inherent property of the hardware at that point? Is it also dependent on the currently loaded bitstream? Is it stored somehow on the device? If the latter, then how is the SUC stored? Flash memory? Fused? For me, this is a major concern with the paper and I do not see it addressed. (Perhaps this was intended to be covered in Section 4, but just doesn’t quite emerge.)

Some new minor editing examples and suggestions.

Line 24 – need an ‘a’ between is and permanent

Line 29 – change “More” to “A more”

Line 31 – replace “generally” with “and other” and then add “devices” at the end of the sentence.

Line 33 – replace “the borne” with “a”

Line 47 – add “an” between “as” and “alternative”

(This list is by no means exhaustive, a thorough review if the revision is still needed.)

Reviewer 2 Report

The work certainly improved. The authors added some relevant information (I appreciate the effort), it is a good work, but connections towards IIoT/IoT and the industry are still not well underlined regarding the proposed concept and alternative approaches already at TRL7-9.
The IoT concept was called 4 times in the paper without any clear indication of a broader view towards applications/products/etc. (it looks to be added only to increase the attractiveness of the paper).

Round 3

Reviewer 1 Report

Revisions to clarify the nature and intent of the SUC generation process are acceptable. Thank you.